# Oral Cavity Cancer Tissues Differ in Isotopic Composition Depending on Location and Staging

**DOI:** 10.3390/cancers15184610

**Published:** 2023-09-18

**Authors:** Katarzyna Bogusiak, Marcin Kozakiewicz, Aleksandra Puch, Radosław Mostowski, Piotr Paneth, Józef Kobos

**Affiliations:** 1Department of Maxillofacial Surgery, Medical University of Lodz, 113 Żeromskiego, 90-549 Lodz, Poland; 2Institute of Food Technology and Analysis, Faculty of Biotechnology and Food Sciences, Lodz University of Technology, 4/10 Stefanowskiego, 90-924 Lodz, Poland; 3Institute of Applied Radiation Chemistry, Lodz University of Technology, 116 Żeromskiego, 90-924 Lodz, Poland; 4Department of Histology and Embryology, Medical University of Lodz, 7/9 Żeligowskiego, 90-752 Lodz, Poland

**Keywords:** oral cavity cancer, tumour, isotopic composition, IRMS, spectrometry, isotopic analysis

## Abstract

**Simple Summary:**

Oral squamous cell carcinomas (OSCCs) pose significant therapeutic challenges. Despite the advancement of treatment methods, they are still characterised by a poor prognosis. The isotopic abundance of oral cancers reflects their biology. A better understanding of OSCC biology may improve treatment outcomes. In this study, the isotopic compositions of the nitrogen and carbon of OSCC tissue samples were investigated across different oral cancer localisations. The correlations between the isotopic composition and clinical and histological advancement were established. The results suggest that assessment of the isotopic composition might serve as a valuable tool for determining tumour biology and behaviour.

**Abstract:**

The aim of this paper was to characterise the isotopic composition of oral squamous cell carcinoma (OSCC) specimens of different areas of the oral cavity. Secondly, we assessed whether there was a correlation between clinical stages of OSCC and isotopic abundance. The IRMS procedure was performed on 124 samples derived from 31 patients with OSCC of 15 N and 13 C to assess the isotopic composition. From each individual, four samples from the tumour, two from the margins, and two samples of healthy oral mucous membranes were derived. The two samples from the tumour and two samples from the margin were additionally subjected to histopathological assessment. Then, statistical analysis was conducted. Tumour infiltration tissues of the lower lip were characterised by higher mean δ^13^C values compared to samples derived from cancers of the other oral cavity regions (−23.82 ± 1.21 vs. −22.67 ± 1.35); (*p* = 0.04). The mean percentage of nitrogen content in tumour tissues was statistically higher in patients with the most advanced cancers (11.89 ± 0.03%) versus the group of patients with II and III stage cancers (11.12 ± 0.02%); (*p* = 0.04). In patients at stage IV, the mean δ^13^C value in the cancer samples equalled −22.69 ± 1.42 and was lower than that in patients at less severe clinical stages (*p* = 0.04). Lower lip cancer tissues differed in the isotopic abundance of carbon in comparison with tissues derived from the group of combined samples from other locations. Values of δ^13^C observed in specimens derived from lower lip cancers were similar to those observed in healthy oral mucous membranes. Cancer tissues obtained from patients in the last stage of OSCC had a different isotopic composition in comparison with those obtained from earlier stages. To confirm these observations, further research on larger groups of patients is needed.

## 1. Introduction

Isotope ratio mass spectrometry (IRMS) is a specialised analytical technique which, based on the analysis of the isotope composition of various tissues, can provide information about the metabolism of the examined tissues. Disorders of cell metabolism, which accompany various disease processes, lead to changes in metabolic pathways and, consequently, to changes in the isotopic composition of cells and tissues [1]. IRMS has been used to assess the isotopic composition of cells from postoperative sections, hair, and plasma in the course of various diseases.

In breast cancer, using cultured cells with the propensity to be invasive, it has been found that metabolic changes appear in the urea cycle, glycolysis, lipid synthesis, and anaplerosis. Moreover, this metabolic shift was related to changes in isotopic composition; they were ^15^N-depleted and ^13^C-enriched [2,3]. In ovarian cancer, where the test material was plasma, researchers noted changes in Cu metabolism and lower δ^65^Cu values in plasma than in healthy controls [4]. A study of bladder cancer patients revealed that differences in the abundance of ^15^N and ^13^C can be used as potential markers to predict the recurrence of the disease. It was demonstrated that ^13^C depletion in normal urothelium in patients with surgically treated bladder cancer is associated with shorter disease-free survival [5].

Research on tumours affecting infants and children, such as neuroblastomas and Wilm’s tumours, revealed that the δ^15^N values in affected tissues were higher compared to normal kidney cortex tissue, which was used as a control. In tissues biopsied from neuroblastomas and Wilm’s tumours, changes in amino acid metabolism were noted [6,7].

Alterations in isotopic composition were also observed in metabolic disorders. Based on tested hair samples, changes in amino acid metabolism were observed in people suffering from anorexia nervosa, nutritional stress, and chronic malnutrition. Increases in δ15N values in hair showed the involvement of metabolic pathways leading to ^15^N-enriched amino acids [8,9]. On the other hand, a higher body mass index was associated with an increase in δ^13^C in hair, reflecting the role of proteins in the diet [8]. δ^15^N in hair in patients with metabolic syndrome has been shown to increase while the value of δ13C decreases [10].

It was also noted that, since the liver is the major site of ammonium acid metabolism, changes in δ15N values can be used to predict the impact of metabolic diseases on liver function, e.g., in patients with cirrhosis of the liver [11].

Studies have also confirmed changes in the metabolism of oral cancer cells, revealing that oncogenic signalling pathways directly promote metabolic reprogramming to increase the biosynthesis of lipids, carbohydrates, proteins, DNA, and RNA, leading to increased tumour growth [12].

Our first study showed that tumour samples of oral squamous cell carcinomas had a decreased δ^15^N value and an increased δ^13^C value when compared to healthy tissue samples.

δ^15^N and δ^13^C values slightly differ between tissues taken from the margin and distant oral mucosa. We also noted that δ^13^C in tumours is positively correlated with alcohol consumption and angioinvasion and inversely related to the BMI index [13].

Despite advancements in the treatment of malignant tumours, the outcome and prognosis for patients with OSCC are still poor. Factors attributed to unfavourable treatment outcomes range from demographic features, clinical factors related to the patient’s general condition (concomitant diseases and oral hygiene), the tumour itself (anatomic site, stage, grade, histopathological features, molecular markers), and the treatment (type and timing of treatment, adjuvant therapy). Demographic indicators include age, gender, ethnic group, social and economic status, and health-related habits (like smoking and alcohol intake). In Eastern Europe, one of the leading contributors to significant mortality and morbidity rates is the advanced clinical stage of patients with oral cancers due to late referral of patients for medical advice. The advancement of oral cancers affects not only the method of surgical treatment but also the use of adjuvant therapy and is related to patients’ general health (nutritional status, anaemia). Another problem is the large heterogeneity in the biology of OSCC. In daily practice, we observe patients with small primary tumours and with very advanced lymph node metastases, as well as patients with huge primary tumours that metastasise slowly. The reasons for the variable biological behaviour of OSCCs remain uncertain. It can be hypothesised that oral cancer biology may be related to their isotopic composition [14,15,16].

A better understanding of the molecular mechanisms affecting oral carcinogenesis could lead to new diagnostic and therapeutic approaches and improve prognosis. We hope this research will contribute to an ongoing discussion regarding the possible utility of isotopic composition assessment as a tool to predict tumour biology and the likelihood of aggressive behaviour. To achieve these objectives, we assessed the isotopic composition of oral squamous cell cancer (OSCC) specimens of different areas of the oral cavity. Secondly, we searched for the correlation between the clinical stages of OSCC and isotopic abundance.

## 2. Materials and Methods

### 2.1. Characteristics of the Study Group and Sample Collection

A total of 31 patients with oral squamous cancer, confirmed by pre-operative histopathological examination, were enrolled in the study. The analysed group consisted of 11 women and 20 men aged 43 to 80 years old. The mean age of the female patients was 68.64 ± 5.82, and that of the male group was 64.55 ± 10.41. All patients from whom samples were taken were from the Lodz voivodeship in central Poland, which means that significant differences in the isotopic composition of their tissues were unlikely. Exclusion criteria for participation in this study were recurrence of the oral cancer and previous and pre-operative chemotherapy and radiotherapy.

Among all patients, 16 were smokers (7 females and 9 males). In this study group, the most common localisations of oral cancers were floor of the mouth (9 patients) and tongue (8 patients). These two groups of patients constituted more than half of the total participants. The least frequent area of primary tumour was the lower lip; only 3 patients had a primary tumour originating in the lower lip. The largest group of patients presented with advanced cancers in stages III and IV. Detailed epidemiological and clinical data are presented in Table 1.

In total, 10 samples were derived from each patient during the surgery: four samples from the tumour, two from the margins, and two samples of healthy tissue. Specimens of healthy oral mucous membranes constituted a reference group. They were obtained from oral mucous membrane at least 40 mm away from the primary tumour border and presented no visible lesion or discolouration. The two tumour sections and two margin sections were flooded with formalin solution and embedded in paraffin. Subsequently, they were subjected to histopathological evaluation, which was performed by an experienced pathomorphologist (JK). The remaining six samples were frozen and stored at −70 °C until they were subjected to further preparatory procedures and IRMS.

Bioethics Committee consent (RNN/185/18/KE) was obtained to conduct this study.

### 2.2. IRMS Procedure

Four specimens derived from each patient were used for IRMS. An assay of the stable δ^15^N and δ^13^C isotopes was performed using the 12 samples. The obtained tissue sections were frozen at −70 °C for another 48 h and freeze-dried (lyophiliser Christ Delta 1–24 LSC, GmbH, Osterode am Harz, Germany).

For the IRMS material, approximately 3 ± 1 mg was weighed out into tin capsules, vanadium pentoxide was used as the combustion catalyst, and the home standard was thiobarbituric acid (calibrated against atmospheric nitrogen and PDB for δ^15^N and δ^13^C, respectively). Analysis of the stable isotopes δ^15^N and δ^13^C was performed by means of a Sercon SL20–22 Continuous Flow Isotope Ratio Mass Spectrometer connected to a Sercon SL elemental analyser for simultaneous carbon–nitrogen analysis. The variation in isotopic composition of nitrogen and carbon elements was measured as isotopic ratios of heavier to lighter isotopes in relation to international standards. The results were expressed in δ values. For carbon isotopes, we obtained the δ^13^C values by comparing the ^13^C/^12^C ratio in the analysed sample to the isotopic ratio ^13^C/^12^C ratio in the international PDB standard (Pee Dee Bee, Belemnitella Americana). For nitrogen isotopes, we used δ^15^N values related to the ^15^N/^14^N ratio of the standard, which is the atmospheric nitrogen. In general, δ values can be described as follows:δX(‰) = (R sample/R standard − 1) × 1000
where δX represents δ^15^N or δ^13^C, and R is isotope ratio of heavier to lighter isotope (^15^N/^14^N or ^13^C/^12^C).

In this study, we also analysed other IRMS values, including the minimal (Min) and maximal (Max) percentage mass content of carbon isotopes (C) and nitrogen isotopes (N); median and interquartile range (IQR); mean ± standard deviation (SD); percentage mass content of C and N; and total nitrogen-to-carbon ratio ([N]/[C]).

The corresponding author (KB) was present during all the surgeries when samples were collected. The corresponding author took all the samples. Then, all samples were carefully labelled with a sticker with information on the patient ID, date of the surgery, and type of sample (tumour 1, tumour 2, margin 1, margin 2, etc.). Next, the samples were segregated, and a list of patients and their samples was made. Samples that were subjected to histopathological assessment and used for the IRMS procedure were transferred together with the list of samples and information from the stickers.

In all tables where IRMS parameters are included, we used average values for each IRMS parameter. We observed incredibly small differences in the values of the parameters of replicates. The samples used for the IRMS procedure did not differ significantly in size. All of them were prepared according to a standardised protocol used for IRMS, and they were weighed before being used for further procedures.

### 2.3. Statistical Analysis

Statistical analysis was conducted to determine any significant differences between the analysed data. Using the Shapiro–Wilk test, the normality of the data distributions was evaluated. The Levene test was used to assess the homogeneity of variance. In cases of a normal distribution, the *t*-test for independent samples or one-way analysis of variance (ANOVA) was used. In other circumstances, the Mann–Whitney test or Kruskal–Wallis test was performed. As a level of significance, a *p* value of <0.05 was determined. All statistical analyses were performed using Statistica Software (version 12), StatSoft, TIBCO Software, Dell, Tulsa, OK, USA.

## 3. Results

Among the patients included in the study, cancer of the floor of the mouth was the most commonly observed, and cancer of the lower lip was the least frequently diagnosed. In the histopathological evaluations, T3 tumour size was dominant, which meant that the analysed tumours were larger than 4 cm and/or had a depth of invasion greater than 10 mm. In this study, patients with advanced cancers prevailed (in stages III and IV of clinical advancement). Most were in stage IV—15 cases (11 men and 4 women), representing almost half of all cases.

The isotope ratio abundance of nitrogen and carbon was investigated in all derived sections obtained intraoperatively with the use of IRMS. The tissues of tumour infiltration differed in their carbon and nitrogen isotopic composition from the healthy oral mucosa. All analysed differences in IRMS measurements proved to be statistically significant (Table 2). The mean percentage mass content of nitrogen (x[N]) in the tumour samples was higher than that in the healthy tissue, 11.50 ± 0.03% vs. 9.52 ± 0.03% (*p* = 0.01). An inverse observation applied to the mean carbon content (x[C]), which was higher in healthy tissue than in tumour tissue, 53.51 ± 0.07% vs. 48.79 ± 0.06% (*p* = 0.001). The mean values of δ^15^N were higher in healthy tissues (9.76 ± 0.72) than in tumour tissues (8.78 ± 0.73) (*p* = 0.000002). In contrast, the mean δ^13^C values appeared to be higher in tumour samples (−22.96 ± 1.39) than in the reference group (−23.696 ± 1.57) (*p* = 0.01).

A search for the relationship between the clinical stage of the primary tumour and IRMS measurements showed that only two parameters had a significant correlation with OSCC staging (Table 3). Due to small subgroup sizes, the patients at stages III and II were combined. The mean percentage of nitrogen content in the tumour tissues was statistically higher in patients with the most advanced cancers (11.89 ± 0.03%) versus the group of patients at stages II and III (11.12 ± 0.02%) (*p* = 0.04). In patients at stage IV, the mean δ^13^C value in the cancer samples was −22.69 ± 1.42, which proved to be statistically significantly lower than in patients at less severe clinical stages (*p* = 0.04). We did not observe any significant correlation between primary tumour size (pT) and analysed IRMS measurements nor between cancer grade and IRMS parameters.

Analysis of the relationship between the location of the primary tumour and the spectrometry measurements showed that tumour infiltration tissues of the lower lip were characterised by higher mean δ^13^C values compared to samples derived from cancers of the other oral cavity regions (−23.82 ± 1.21 vs. −22.67 ± 1.35); (*p* = 0.0402). Detailed data are presented in Table 4.

Of the unfavourable prognostic factors present in the histopathological examination of OSCC sections, only two features were found to have correlations with IRMS measurements (Table 5). These were the presence of lymph node capsular infiltration and neuroinvasion. The presence of nodular capsule infiltration was associated with a higher mean percentage mass content of nitrogen (12.9 ± 0.01 versus 11.15 ± 0.03) (*p* = 0.02) and a higher total nitrogen-to-carbon ratio (0.28 ± 0.01 versus 0.23 ± 0.07); (*p* = 0.01). The presence of neuroinvasion was associated with a higher mean δAir value (9.16 ± 0.43 vs. 8.66 ± 0.75) (*p* = 0.04).

Both ENE(+) and ENE(−) tumours exhibited nitrogen levels higher than those in healthy tissue, which was 9.52%, indicating a definite upward tendency in comparison to normal tissues.

## 4. Discussion

Analysis of our demographic data revealed that the majority of patients included in this study were at stages III and IV of oral cancer. According to statistics, around 30–50% to as many as two-thirds of cases of OSCCs are discovered at an advanced stage. Higher rates of late diagnosis are found in Eastern Europe, while a lower incidence of oral cancers at stages III and IV is observed in the United States [17,18]. In our study, patients with cancers originating in the floor of the mouth prevailed (nine patients; fraction 0.29). The least common localisation of oral cancer was the lower lip (three patients, fraction 0.10). These results were in accordance with the data reported by other authors (22% in the floor of the mouth; lower lip is often included in the “remaining sites” category, below 4%) [19].

Our study confirmed that the isotopic composition of tissues derived from oral cancers differed significantly from the reference group (healthy oral mucous membrane). It was observed that, in tumour specimens, the average nitrogen content was higher (12.49% vs. 9.78%; *p* = 0.0127), and the average carbon content was lower in comparison to the reference group (46.39% vs. 52.25%; *p* = 0.0012). We also noted that samples of cancerous infiltration were characterised by lower mean values of δ^15^N (8.78 ± 0.73 vs. 9.76 ± 0.72; *p* = 0.0000), i.e., by a depletion of the 15N isotope. We also discovered that oral cancer tissues had higher mean values of δ^13^C (−22.69 vs. −23.62; *p* = 0.0156), i.e., higher content of the 13C isotope. Some authors who have conducted investigations on malignant tumours of other parts of the body have obtained similar results. The results of in vivo research on breast cancer cultured cells confirmed that metabolic changes related to tumour invasiveness presented as 15N depletion and 13C enrichment [2,3].

Our study also revealed that tissue samples of oral cancers derived from patients at stage IV of clinical advancement differed in isotopic composition from those obtained from individuals with lesser advancement of OSCC (patients at clinical stages II and III). The progression of cancer can impact the isotopic composition of tissues and metabolic processes within the cancer cells. In our study, in tumour specimens from patients with the most advanced cancers, the mean percentage of nitrogen content (12.57 vs. 12.42; *p* = 0.0432) and mean δ^13^C values were significantly higher (−22.60 vs. −23.32; *p* = 0.0412). This ^13^C enrichment in samples obtained from loco-regionally advanced oral cancers may reflect alterations in the metabolic pathways that occur in cancer cells during tumour progression. One of the key changes that occurs in cancer cells as the disease progresses is an increase in the Warburg effect, which is characterised by a shift towards glycolysis and away from oxidative phosphorylation. This shift results in an increase in the production of lactate and a decrease in the level of adenosine triphosphate (ATP), which is the primary source of energy for cells. The expression and activity of numerous enzymes involved in metabolism, including those involved in amino acid metabolism and fatty acid metabolism, can change according to the loco-regional advancement. It has been confirmed that metabolic pathways operate at different speeds at different stages of oncogenesis [20]. The changes in the isotopic content of nitrogen and carbon can provide insight into the cellular metabolism of tissue samples. It has been shown that stable isotopes of nitrogen (^14^N and ^15^N) and carbon (^12^C and ^13^C) can be used to trace the functioning of metabolic pathways [1]. The relative abundance of ^15^N in a tissue sample can be used to determine the rate of nitrogen incorporation into cellular proteins and other nitrogen-containing biomolecules. The process of nitrogen incorporation is tightly linked to cellular metabolism, as the rate of nitrogen incorporation is influenced by the rate of protein synthesis, the rate of protein degradation, and the rate of nitrogen turnover in the cell. Similarly, the relative abundance of ^13^C in a tissue sample can be used to assess the rate of carbon incorporation into cellular lipids, carbohydrates, and other carbon-containing biomolecules. The rate of carbon incorporation is influenced by the rate of cellular respiration and the rate of carbon turnover in the cell.

On the basis of the available literature, it can be assumed that changes in the isotopic content of nitrogen and carbon in tissue samples can be used to diagnose and monitor the progression of cancer [5,6,7]. For example, lower levels of ^15^N in tumour tissue compared to healthy tissue can indicate decreased cellular protein synthesis in the tumour. Additionally, changes in the isotopic content of carbon in tumour tissue compared to healthy tissue can indicate changes in cellular respiration and energy metabolism. For example, higher levels of ^13^C in tumour tissue compared to healthy tissue can indicate increased metabolism and cellular respiration of the tumour.

The analysis of our results suggested that samples derived from oral cancers have different nitrogen and carbon isotopic abundance from healthy tissues of the oral mucous membrane. In this study, we observed that tissues obtained from the lower lip differed from samples derived from other primary tumour locations with regard to the isoptic abundance of carbon. Our results suggested that lower δ^13^C values are more characteristic of healthy oral mucosa than cancer tissues. δ^13^C values measured for lower lip cancer tissues were lower in comparison to δ^13^C values obtained from other primary tumour locations.

We assumed that lower lip cancers may be more similar to healthy tissues in terms of the pace of metabolic pathways of cellular respiration, which manifests as similar levels of ^13^C.

It can be hypothesised that lower lip cancers are less aggressive than those derived from the other analysed locations of primary oral cancers. The milder biological behaviour of lower lip cancers may be a result of their metabolism being less altered in comparison to healthy oral mucous membranes, which is reflected in IRMS measurements. Clinical experience confirms that the progress of lower lip cancers is slower; the time to lymph node metastases appearance and further spread of the cancer is longer in comparison with oral cancers of other locations. Thus, the prognosis is better, and the survival rate and time to recurrence is longer. Various studies have confirmed that lower lip OSCCs metastasise at a late stage and have a better prognosis compared to oral cancers of other localisations. Patients with lower lip cancer have a 5-year survival rate of approximately 90%.

In cases with a localised stage of lower lip cancer, the 5-year survival rate is 94% [21]. In contrast, the overall 5-year survival rate for oral cancers (including all localisations and stages) is estimated to be 60% [22]. The literature indicates that the floor of the mouth is the localisation of OSCC with the lowest survival statistics. On average, only 53% of patients with cancer of the floor of the mouth (in the general population, with all stages combined) are still alive 5 years after diagnosis. Overall, the outcome and prognosis for people with OSCC are still poor when considering the general group of patients in all stages and localisations, in adult men and women of all ages. Nevertheless, the primary tumour localisation seems to be an important factor influencing survival rate. Although this study comprised a small number of patients, we believe that our observations are significant. Further studies on a larger group of patients with different localisations and stages, and with at least 5 years of follow-up, are needed to confirm our results and to establish reliable clinical applications. The similar ^13^C abundance in samples from lower lip cancers and healthy mucosa may reflect the similarity of their metabolism. This could explain the less aggressive behaviour and better overall prognosis of this type. It could be hypothesised that lower lip cancers have different biology from other locations of OSCC, and they could evolve from distinct oncogenic processes. This may reflect different tumour metabolism, which depends on clinical advancement. We assumed that the selected patients with lip cancer and δ^13^C values that were similar to the δ^13^C values of the healthy mucous membrane could benefit from less aggressive treatment without compromising survival statistics.

One of the most important negative prognostic factors of OSCC is the presence of metastases in the lymph nodes of the neck. Its appearance usually means a worsening of the overall prognosis and a decrease in the 5-year survival rate to 25–40%, while in patients with no signs and symptoms of cancer spreading to the lymph nodes, 5-year survival is estimated to be 90% [23,24]. The grading, staging, size of the margins of surgical resection, and location and size of the tumour also affect the loco-regional recurrence rate and long-term outcome. Certain studies have concluded that in poorly differentiated carcinomas, metastases and cancer cells are more frequently observed in the margins and are, therefore, characterised by reduced survival rates [25]. In this study, we analysed well-established negative risk factors of histopathological examination of OSCC. We observed that only two of them, namely the presence of lymph node capsular infiltration and neuroinvasion, were related to IRMS parameters measured in the tumour samples. Capsular infiltration of lymph nodes was reflected in a higher mean percentage mass content of nitrogen (12.90 ± 0.01 vs. 11.15 ± 0.03) and a higher total nitrogen-to-carbon ratio (0.28 ± 0.01 vs. 0.23 ± 0.07). Neuroinvasion was characterised by higher mean δAir values. We did not find any scientific literature that specifically connects the concept of negative risk factors of histopathological examination of OSCCs to cancer cell metabolism assessed by IRMS.

The presence of extranodal extension, as well as evidence of neuroinvasion observed in postoperative histopathological examination, is a sign of cancer progression. These are factors related to late clinical stages and distant spread. Upregulation of nitrogen uptake at this point can be the result of dysregulated signalling pathways, such as mTOR, promoting the uptake of nitrogen, combined with compounds like amino acids and nucleotides [26].

It appears that cancer cell metabolism pathways are changed to maximise the use of nitrogen and carbon for the synthesis of different macromolecules and also by reducing catabolic processes and nitrogen disposal.

In addition to their direct effects on cancer cell metabolism, it was concluded that nitrogen-containing compounds play a role in the host response to cancer progression. For example, the immune system relies on nitrogen-containing molecules such as amino acids to produce immune cells and antibodies that can target and eliminate cancer cells. In advanced stages of cancer, the immune system is insufficient, leading to a decrease in immune function and an increase in tumour growth [27].

Moreover, the accumulation of nitrogenous waste products, such as ammonia, has been shown to promote tumour growth and angiogenesis. As cancer cells proliferate, they produce large amounts of waste products that must be detoxified or eliminated. At some point in development, cancer cells may be unable to effectively detoxify ammonia, leading to its accumulation and promotion of tumour growth [28].

Another possible explanation for the increase in nitrogen content in advanced cancer stages is the presence of necrotic tissue. As tumours grow, they can outstrip their blood supply, leading to areas of necrosis or cell death. Necrotic tissue is characterised by the release of nitrogen-containing compounds, such as proteins and nucleic acids, which can contribute to the overall nitrogen content of the tumour. In advanced stages of cancer, the amount of necrotic tissue present is usually higher, leading to an overall increase in nitrogen content.

Furthermore, the increase in nitrogen content in advanced cancer stages may be due to the presence of tumour-associated macrophages (TAMs), which are immune cells that can promote tumour growth and metastasis. TAMs have been shown to have high levels of arginase, an enzyme that converts arginine into ornithine and urea [29].

The increase in nitrogen content in tissue samples derived from the tumours of patients in advanced cancer stages is likely due to a combination of factors related to cancer cell metabolism, host response to cancer progression, and tumour microenvironment. Dysregulated nitrogen uptake and utilisation pathways in cancer cells, the immune system’s response to cancer, the accumulation of nitrogenous waste products, the presence of necrotic tissue, and the activity of TAMs (tumour-associated macrophages) are all potential contributors to the overall increase in nitrogen abundance.

Recent findings support the diagnostic use of nitrogen isotopic analysis in the differentiation of malignant from healthy tissues. The novel analytical method proposed by Straub M. et al. allows for the assessment of the amount of tumour cell infiltration at the tumour margin, which is one of the most significant problems following surgery, as it is related to tumour recurrence [30].

In this study, we did not assess the influence of HPV proteins or genome presence on the IRMS parameters. This will be an interesting topic to investigate since it has been proven that HPV-induced cancers have a different prognosis and response to radio- and chemotherapy. Many studies have confirmed the prevalence of HPV proteins/mRNA/genome in samples derived from oral cancers. To date, the association between HPV infection and OSCCs is still not clear. We believe that HPV is a possible oncogenic factor for oral cancer, but further studies are required to clarify this potential association [31].

## 5. Conclusions

Lower lip cancer tissues differed in their isotopic abundance of carbon in comparison with tissues derived from the group of combined samples from other locations of oral cancers. Values of δ^13^C observed in specimens derived from lower lip cancers were similar to those observed in the healthy oral mucous membrane.

Cancer tissues obtained from patients in the last stage of OSCC had a different isotopic composition in comparison with those obtained from earlier stages.

To confirm these observations, further research on larger groups of patients is needed.

## Figures and Tables

**Table 1 cancers-15-04610-t001:** Demographic data.

		Male	Female	Total
Number of patients		20	11	31
Age (mean)		64.55 ± 10.68 SD	68.64 ± 6.10 SD	66.00 ± 9.26 SD
BMI (mean)		22.76 ± 4.28 SD	27.63 ± 5.16 SD	24.49 ± 5.03 SD
Smoking		9	7	16
Number of cigarettes per day	10–20	1	3	4
	>20	8	4	12
Alcohol consumption		11	1	12
Primary localisation	Tongue	5	3	8
	Floor of the mouth	7	2	9
	Lower alveolar ridge	4	2	6
	Lower lip	3	0	3
	Upper alveolar ridge	1	4	5
pTNM ^1^	pT1	0	0	0
	pT2	6	3	9
	pT3	7	5	12
	pT4	7	3	10
	pN0	10	8	18
	pN1	4	2	6
	pN2	6	1	7
Staging	II	5	2	7
	III	4	5	9
	IV	11	4	15
Grading	G1	5	4	9
	G2	12	6	18
	G3	3	1	4
ENE ^2^ (+)		7	6	13
Angioinvasion (+)		9	2	11
Neuroinvasion (+)		5	2	7

^1^ pTNM: pathomorphological TNM. ^2^ ENE: extranodal extension.

**Table 2 cancers-15-04610-t002:** Comparison of IRMS parameters for tumour tissue and healthy tissue.

		Tumour	Healthy Tissue	*p* Value
Nitrogen	Min-Max	3.15–13.35%	2.66–13.81%	*p* = 0.0127
	Median	12.49%	9.78%	
	IQR ^1^	1.20%	4.09%	
Carbon	Min-Max	44.63–69.49%	44.55–73.73%	*p* = 0.0012
	Median	46.39%	52.25%	
	IQR	2.24%	9.35%	
[N]/[C] ^2^	Min-Max	0.05–0.29	0.04–0.31	*p* = 0.0083
	Median	0.27	0.19	
	IQR	0.03	0.11	
δ^15^N (‰)	Min-Max	7.50–11.28	7.70–11.68	*p* = 0.0000
	Mean ± SD	8.78 ± 0.73	9.76 ± 0.72	
δ^13^C (‰)	Min-Max	−27.30–−20.86	−27.51–−20.29	*p* = 0.0156
	Median	−22.69	−23.62	
	IQR	1.39	0.84	

^1^ IQR: interquartile range. ^2^ [N]/[C]: total nitrogen-to-carbon ratio.

**Table 3 cancers-15-04610-t003:** Correlation between IRMS measurements and tumour clinical advancement.

		Clinical Stage		*p* Value	pT2	pT3	pT4	*p* Value
		II + III	IV					
Nitrogen	Min–Max	6.50–13.35	3.15–13.35	0.0432	7.92–13.00	6.50–13.35	3.15–13.35	0.8731
	Median	12.42	12.57		12.61	12.57	12.57	
	IQR	1.37	0.76		1.19	0.99	3.03	
Carbon	Min–Max	44.63–63.71	44.79–69.49	0.7089	44.63–59.52	44.81–63.71	44.79–69.49	0.7437
	Median	46.44	46.44		46.48	46.44	46.44	
	IQR	2.92	1.46		1.36	2.92	4.42	
[N]/[C]	Min–Max	0.10–0.29	0.05–0.29	0.1844	0.14–0.29	0.10–0.29	0.05–0.29	0.7141
	Median	0.27	0.27		0.23	0.27	0.27	
	IQR	0.03	0.02		0.02	0.03	0.08	
δ^15^N (‰)	Min–Max	7.50–11.28	7.70–10.33	0.8193	7.50–10.33	7.93–11.28	7.81–10.29	0.6474
	Mean ± SD	8.75 ± 0.77	8.81 ± 0.66		n/a	n/a	n/a	
	Median	n/a	n/a		8.63	8.98	8.68	
	IQR	n/a	n/a		1.39	0.95	1.22	
δ^13^C (‰)	Min–Max	−27.30–−21.95	−26.60–−20.79	0.0412	−27.30–−21.38	−25.91–−21.45	−26.60–−20.79	0.7788
	Median	−23.32	−22.60		−22.88	−23.11	−22.86	
	IQR	2.03	1.79		1.13	2.38	2.46	

**Table 4 cancers-15-04610-t004:** Correlation between IRMS measurements and primary tumour location.

		Lower Lip	Other Localisation	*p* Value
Nitrogen	Min–Max	6.64–13.35	3.15–13.35	0.8900
	Median	12.19	12.49	
	IQR	6.71	0.97	
Carbon	Min–Max	45.87–63.71	44.63–69.49	0.2399
	Median	48.75	46.39	
	IQR	17.84	1.36	
[N]/[C]	Min–Max	0.10–0.29	0.05–0.29	0.6783
	Median	0.25	0.27	
	IQR	0.19	0.02	
δ^15^N (‰)	Min–Max	8.02–11.28	7.50–10.33	0.1234
	Mean ± SD	9.39 ± 1.11	8.71 ± 0.62	
δ^13^C (‰)	Min–Max	−25.05–−21.95	−27.30–−20.79	0.0402
	Median	−23.77	−22.87	
	IQR	2.67	1.08	

**Table 5 cancers-15-04610-t005:** Correlation between IRMS measurements and tumour characteristics in histopathological examination.

		Histopathological Stage		*p* Value	pN		*p* Value	ENE ^1^		*p* Value	Angioinvasion		*p* Value	Neuroinvasion		*p* Value
		G1 + G2	G3		pN 0	pN (+)		ENE(+)	ENE(−)		Yes	No		Yes	No	
Nitrogen	Min–Max	6.50–13.35	3.15–12.67	0.1896	6.50–13.35	3.15–13.35	0.2804	12.42–13.35	3.15–13.35	0.0183	3.15–13.35	6.50–13.35	0.8088	3.15–13.00	6.50–13.35	0.9219
	Mean	n/a	n/a		n/a	n/a		12.90 ± 0.01	11.15 ± 0.03		n/a	n/a		n/a	n/a	
	Median	12.61	12.42		12.28	12.72		n/a	n/a		12.69	12.42		12.42	12.57	
	IQR	0.94	7.66		0.90	2.19		n/a	n/a		3.16	0.96		5.06	0.97	
Carbon	Min–Max	44.63–63.71	45.38–69.49	0.4829	44.63–63.71	45.11–69.49	1	45.24–46.84	44.63–69.49	0.2040	45.11–69.49	44.63–63.71	0.7414	44.79–69.49	44.63–63.71	0.5896
	Median	46.39	46.29		46.45	46.29		45.87	46.48		46.26	46.41		46.24	46.45	
	IQR	1.37	18.56		1.99	2.44		1.12	2.78		4.04	1.84		14.41	1.56	
[N]/[C]	Min–Max	0.10–0.29	0.05–0.27	0.2342	0.10–0.29	0.45–0.29	0.1966	0.27–0.29	0.05–0.29	0.0119	0.05–0.29	0.10–0.29	0.5824	0.05–0.28	0.10–0.29	0.8063
	Median	0.27	0.27		0.27	0.28		0.28	0.27		0.28	0.02		0.27	0.27	
	IQR	0.02	0.19		0.03	0.06		0.01	0.05		0.08	0.02		0.14	0.03	
δ^15^N (‰)	Min–Max	7.50–11.28	7.81–10.29	0.9860	7.5–11.28	7.7–10.33	0.6623	7.7–10.33	7.5–11.28	0.5663	7.81–10.29	7.5–11.28	0.7473	8.38–10.29	7.5–11.28	0.0444
	Mean	8.78 ± 0.74	8.78 ± 0.57		8.85 ± 0.61	8.73 ± 0.78		8.62 ± 0.72	8.82 ± 0.71		8.72 ± 0.53	8.81 ± 0.80		n/a	n/a	
	Median	n/a	n/a		n/a	n/a		n/a	n/a		n/a	n/a		9.10 ± 0.43	8.68 ± 0.81	
	IQR	n/a	n/a		n/a	n/a		n/a	n/a		n/a	n/a		1.01	0.98	
δ^13^C (‰)	Min–Max	−27.30–−21.33	−26.60–−20.79	0.8309	−27.3–−20.79	−26.6–−21.38	0.2804	−23.43–−21.38	−27.3–−20.79	0.1123	−26.98–−20.79	−27.3–−21.33	0.4679	−26.42–−25.97	−25.91–−20.79	0.8831
	Mean	n/a	n/a		n/a	n/a		−22.16 ± 0.61	−23.17 ± 1.42		n/a	n/a		n/a	−22.86 ± 1.17	
	Median	−22.90	−23.33		−22.8	−23.06		n/a	n/a		−22.84 ± 1.65	−23.02 ± 1.18		−23.28 ± 1.75	−22.86 ± 1.19	
	IQR	1.13	4.54		1.10	1.79		n/a	n/a		2.39	0.83		3.68	1.28	

^1^ ENE: extranodal extension (+)—present; (−)—not observed.

## Data Availability

The data on which this study is based will be made available upon request at https://www.researchgate.net/profile/Marcin-Kozakiewicz (accessed on 1 January 2023).

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
