# Peer review of "Oral Cavity Cancer Tissues Differ in Isotopic Composition Depending on Location and Staging"

_cancers, 2023, doi:10.3390/cancers15184610_

Round 1

Reviewer 1 Report

 This study characterises the isotopic composition of oral squamous cell cancer (OSCC) specimens of different areas of the oral cavity. The authors alos ssessed if there was a correlation between clinical stages of OSCC and isotopic abundance. The topic is interesting, while some concerns must be addressed appropriately.

1. Line 97-101 should be deleted. 

2. Some tables should be presented as Figures to improved the readability of this studies. 

3. In the conclusion, the authours claimed that " There are many causes standing for this situation. The leading contributor is late referral of patients for medical advice. Another problem constitutes a large heterogeneity of biology of OSCC". However, "the leading contribution" may not be solved by the findings of this study. Thus, the introduction and disscussion should be modified to be more suitable for present study.

4. The prognosis of lip cancer is usually good. One of important finding of present study is that "Lower lip cancer tissues differed in isotopic abundance of carbon in comparison with other analysed in this study locations of oral cancers". What is the clinical significance of this finding? Especially the authors claimed that "the outcome and prognosis for patients with OSCC are still poor". It seems contradictory. This must be addressed appropriately.

Minor editing of English language required.

Author Response

Dear Sir/Madame,

I hope this email finds you well. I am writing to express my sincere gratitude for your time and invaluable feedback on our article, which we recently submitted for  the review.

I am pleased to inform you that we have carefully considered all of your suggestions and most of them we have incorporated into the revised version of the article. However, we would like to discuss a few of them. In addition, we have made some linguistic corrections and we have checked our text for typing errors.

We hope that the revised version of the article meets your expectations. If you have any further suggestions or comments, we would be honoured to consider any additional input you may have.

With sincere gratitude,

Authors

Response to Reviewer 1

  1. The text of lines 97-101 was rearranged.
  2. We could not manage to present the results in the form of a figure due to the large amount of data. We tried to introduced your suggestion to improve the readability of the results. We have rearranged the content of the tables. We hope it improved the clarity of the text.
  3. The paragraph concerning factors attributed to unfavourable prognosis of patients with OSCC was rewritten. Some additional information was added.
  4. We have introduced some corrections to the text concerning prognosis of oral cancers and lower lip cancers. We hope that you will find our text more clear and properly addressed

Reviewer 2 Report

In the manuscript by Bogusiak et al the authors describe a study of OSCC elemental composition. Isotope ratio mass spectrometry was used to measure the 13C/12C and 15N/14N ratios in multiple samples (tumor, margins, healthy tissue) from 31 patients making the study appear to be robust.

However, the study suffers from some inadequacies in the methods descriptions requiring significant revision. Margins samples were supposedly analyzed but never shown. It is unclear how replicates were considered. It is unclear which tests were used where. Tables are poorly made with many unnecessary rows which altogether detracts from the message.

Detailed comments are below.

P3 Table 1 could benefit from a total column.

P4 L123  the authors specify that 10 samples per patient were obtained, 4 tumor + 4 margins + 2 healthy. Out of those, 2 tumor and 2 margin samples were made into FFPE blocks while the remaining 6 were stored and later used for IRMS (P4L137) or other procedures.

However, Figure 2 lists only tumor and healthy tissue and makes no mention of having replicates. If indeed margin tissue was not evaluated at all and if only 1 sample was used for IRMS, then the manuscript should be rewritten to omit the obsolete information that only confuses the readers.

P4 L 135 IRMS procedure is a bit unclear. The authors thoroughly describe the “delta” measurements. However, the table 2 presents data for “Nitrogen” and “Carbon” which is very different to align with the expected data as described in the methods section. Materials and results sections should be rewritten to match in substance. Ie. P5L161 gives these measures as a side note while they represent the main bulk of results

P5 L 167 the statistics section doesn’t elaborate on how the replicates were considered. Ie. if 2 tumor samples were measured for each patient, was the average used when making Table 2, were some measures outliers? Considering replicates as separate samples might inflate te sample size and lead to erroneous p value estimation. What happened with margin samples? Were paired tests considered given that tumor and healthy tissue are matched from the same patient?

P5 L 167 The authors specify that normality assessment of the data was performed. However, the readers are not shown those results. This makes it difficult to know whether mean+-SD is an appropriate measure for summarizing the resulting data or not.

P5 Table 2. If the authors feel that min and max measures are needed, consider using range (min-max) instead of 2 separate rows. It would be preferable to report only Mean+-SD (if normal) or Median and IQR (for not normal distribution) and make the tables concise. Detailed numbers can be reported in supplement but currently the manuscript tables waste a lot of space needlessly on something that is not referred to or used in the rest of the document. The methods imply t test of MannWhitney test was used but it is not specified which test does the p value come from in the table. P values should be rounded to 4 decimal places, 6 seems excessive.

P6 Table 3 there appears to be a typo in nitrogen pT4 measurement. It is unlikely that the mean is 1.05 and median 12.57. Min and max as separate rows are also superfluous. +-SD in the 1st column is erroneous since SD applies only to some rows.

Did gender have a role in element abundance? Since this factor was used in table 1 implying its importance in clinics it is unusual to see it not used elsewhere.

P7 Table 4 is mostly meaningless since there are only 3 cases with lower lip out of 31 patients. It might be more meaningful to present all different localizations (with grouping) and show ANOVA/kruskal wallis  p values when presenting localization differences. The p value shown is likely and artifact or false positive association due to many individual tests being done.

P8 L218 the additional PHD parameters shown here were not specified as being measured/obtained in the materials section, nor were they presented in table 1. Previous sections should be revised to include these parameters as well. Currently it is impossible to know sample numbers and thus whether the data on table 5 makes sense or not.

P10 Discussion has wrong page numbering

P10 L246 it would be beneficial for the authors to report both their obtained values as well as those reported in the references to put their results in context.

Not much is given about what can lead to depletion of N15 or C13 as well expected size of the change. Without such background or discussion it is very difficult to conclude dPDB measurement of -22.96 +-1.39 would be clinically meaningful compared to -23.73+-1.56 despite the nominal p=0.0156

P11 L 293 The discussion about  lower lip cancers cannot really be supported by the data since there were only 3 cases of lower lip

P11 L 320-331 the discussion (and possibly results) about ENE could be possibly improved by highlighting that while ENE+ cancers had 12.9% nitrogen and ENE- ones had 11.15 it was both higher than in healthy tissue which had 9.52% showing a clear trend of increase from normal. However, since the authors do not specify the number of ENE+ cases it is difficult to say whether the 12.9% number is robust or not.

some typos to be fixed

Author Response

Dear Sir/Madame,

I hope this email finds you well. I am writing to express my sincere gratitude for your time and invaluable feedback on our article, which we recently submitted for  the review.

I am pleased to inform you that we have carefully considered all of your suggestions and most of them we have incorporated into the revised version of the article. However, we would like to discuss a few of them. In addition, we have made some linguistic corrections and we have checked our text for typing errors.

We hope that the revised version of the article meets your expectations. If you have any further suggestions or comments, we would be honoured to consider any additional input you may have.

With sincere gratitude,

Authors

Response to Reviewer 2

Margin samples were analysed but were never shown

P3. According to your suggestions we made the necessary modifications to the table 1 to include the total column.

P4 L123. Text of material and methods section concerning of margin samples was modified. In this research margin tissue samples were used only for histopathological assessment.

P4 L123. To make the text of material and methods coherent with results section we described  parameters used in tables

P5.  In this research the corresponding author (KB) was present during all the surgeries when samples were collected, because she deals with almost all oncological surgeries at the Department.  The corresponding author took by herself all the samples. Then, all samples were carefully labelled – sticker with information concerning patients’ ID, date of the surgery, type of the sample (tumour 1, tumor 2, margin 1, margin 2, etc.) was put on each sample respectively. Next, samples were segregated, and a list of patients and their samples was made. Samples that were subjected to histopathological assessment and for IRMS procedure were transferred together with the list of samples and information from the stickers.

We believe that our labelling system excluded the possibility of mistake i.e. that it is impossible that samples derived from tumour and  healthy tissues did not match from the same patient.

In table 2 and in all tables where IRMS parameters were included we used average values for each IRMS parameter. We observed incredibly small differences in values of parameters of replicates. The samples for IRMS procedure did not differ significantly in their size. All of them were prepared according to standard procedure used for IRMS. They were weighted.

Section of material and methods was updated with above mentioned information.

P5 Table 2 All tables were rearranged to present normality assessment - mean±SD was used for parameters where normal distribution assessment was present and Mean value + IQR were used in case of lack of normal distribution.

Range (min-max) was used instead of 2 separate rows and p values were rounded to 4 decimal places.

P6 Table 3 The typo in Table 3 was corrected

P6. There are a lot of factors affecting carbon and nitrogen abundance in tissues. It is believed that the greatest impact has patients’ nutritional and physiopathological condition.

The topic of isotope abundance diversity according to gender is a very interesting but it requires conducting a research on a large group of individuals to make a reliable conclusion.

In this article we presented epidemiological data in relation to gender, like it is usually done in articles concerning head and neck cancers

As the gender is indeed an important factor we have added to the table 1 more parameters (ENE, Angioinvasion and Neuroinvasion). We hope that it makes data more clear.

Ad P7 and P11 L293. According to our research lower lip cancers have metabolic pathways of cellular respiration that are more similar to healthy tissues. They are also less aggressive and have better 5-year survival rate then cancers in other head and neck locations. We believe that patients with lip cancer who exhibit δPDB values similar to those of healthy mucous membrane could potentially benefit from less aggressive treatment without compromising survival statistics. However we are sure that further studies involving larger patient groups and longer follow-up periods are needed to establish reliable clinical applications.

P8 L218 Additional parameters were presented in table 1

P10 Page numbering was corrected.

Ad P10 L246 As it was suggested, where it was possible the values from the references were mentioned in the text to present the broader context of results obtained by the authors

There is no data concerning how the size of the change of the depletion of  N15 or C13 is related to clinical findings. Unfortunately these variations of abundance of these isotopes are still poorly understood. We believe that experimental researches and that based on theoretical models are needed to better understand all the aspects of changes of isotope abundance and to explain underlaying mechanisms.

Ad P11 L320-331 As it was mentioned by the reviewer, both ENE+ and ENE- tumours exhibited nitrogen levels higher than healthy tissue – this information was included to the text.

In the whole text “Δ” was changed to ”δ”

Reviewer 3 Report

The authors describe the isotopic composition of OSCC patients in tumor, margin and healthy tissue. 186 samples from 31 patients in mostly advanced states were investigated. Overall the authors have investigated an interesting project however I have a few remarks: 

- what impact has their finding on the survival or recurrence of the patients? Can you present follow-up data?

- what is the impact of the location the samples were collected? In the discussion the authors briefly mention the impact of tumornecrosis. Are the findings just related to intratumoral microinvironment?

- Can you comment on other risk factors of OSCC such as HPV16. Did they collect these data?

- Is there a clinical relevance in diagnosis or a therapeutical implication?

General aspects: 

Introduction is lengthy to read, the method has to be better described. The abstract and conclusions section has to be rewritten as the proposed abstract does not clearly describe what the authors conducted and the  conclusion is not backed by the findings in the text. 

Annoying are many small typing errors throughout the text. 

Author Response

Dear Sir/Madame,

I hope this email finds you well. I am writing to express my sincere gratitude for your time and invaluable feedback on our article, which we recently submitted for  the review.

I am pleased to inform you that we have carefully considered all of your suggestions and most of them we have incorporated into the revised version of the article. However, we would like to discuss a few of them. In addition, we have made some linguistic corrections and we have checked our text for typing errors.

We hope that the revised version of the article meets your expectations. If you have any further suggestions or comments, we would be honoured to consider any additional input you may have.

With sincere gratitude,

Authors

Response to Reviewer 3

- what impact has their finding on the survival or recurrence of the patients? Can you present follow-up data?

That's a very interesting point but unfortunately we don't have a 5 year follow up yet. We believe this topic is worth investigating once further research results are available. We plan to continue our research

- what is the impact of the location the samples were collected? In the discussion the authors briefly mention the impact of tumornecrosis. Are the findings just related to intratumoral microinvironment?

Indeed in discussions we mentioned that tumour necrosis affects the tumour tissues’ metabolism.

The tumour samples subjected to IRMS had the parallel samples that were submitted to the histopathological assessment to obtain all histopathological features used in the study and also to exclude the presence of tumour necrosis. We believe that our findings are related to intratumoral microenvironment.

- Can you comment on other risk factors of OSCC such as HPV16. Did they collect these data?

Unfortunately we routinely did not perform IHC p16 assessment so we did not enclose this data into the study.

We did not assessed the presence of the E6 and E7 mRNA in tumour samples

- Is there a clinical relevance in diagnosis or a therapeutical implication?

According to our research lower lip cancers have metabolic pathways of cellular respiration that are more similar to healthy tissues. They are also less aggressive and have better 5-year survival rate then cancers in other head and neck locations. We believe that patients with lip cancer who exhibit δPDB values similar to those of healthy mucous membrane could potentially benefit from less aggressive treatment without compromising survival statistics. However we are sure that further studies involving larger patient groups and longer follow-up periods are needed to establish reliable clinical applications.

Introduction is lengthy to read, the method has to be better described. The abstract and conclusions section has to be rewritten as the proposed abstract does not clearly describe what the authors conducted and the  conclusion is not backed by the findings in the text.

Unfortunately due to other reviewers’ comments the new version has even longer introduction. The method section was enriched in some essential information. The abstract and conclusion section was rewritten.

Round 2

Reviewer 2 Report

Apparently most of the issues were addressed.

The authors appeared to misunderstand tye question about paired tests where i was referring to the statistical tests while the authors elaborated on the actual sample collection procedures.

Unfortunately there is no clean manuscript text and since im on my vacation i cannot readily see the what the final tables would look like on the phone. Proofreading step should be thorough

Author Response

Dear Sir/Madame,

I hope this email finds you well. I am writing to express my sincere gratitude for your time and invaluable feedback on our article, which we recently submitted for the review.

I apologize that we missed the answer on one of your qustions. I hope this time we will meet your expectations.

If you have any further suggestions or comments, we would be honoured to consider any additional input you may have.

With sincere gratitude,

Authors

Response to the reviewer

In table 2 the averages were used. The measures did not outlier significantly. We observed only small differences. We used averages to minimize the impact of sample size on statistical analysis. We believe that weight of sample should not be considered as a confounding factor – before IRMS procedure all samples were weighted.